# The Memristive Properties and Spike Timing-Dependent Plasticity in Electrodeposited Copper Tungstates and Molybdates

**DOI:** 10.3390/ma16206675

**Published:** 2023-10-13

**Authors:** Dawid Przyczyna, Krzysztof Mech, Ewelina Kowalewska, Mateusz Marzec, Tomasz Mazur, Piotr Zawal, Konrad Szaciłowski

**Affiliations:** 1Academic Centre for Materials and Nanotechnology, AGH University of Science and Technology, Mickiewicza 30, 30-059 Krakow, Poland; dawidp@agh.edu.pl (D.P.); kmech@agh.edu.pl (K.M.); kowalews@agh.edu.pl (E.K.); marzecm@agh.edu.pl (M.M.); tmazur@agh.edu.pl (T.M.); piotr.zawal@gmail.com (P.Z.); 2Faculty of Physics and Applied Computer Science, AGH University of Science and Technology, Mickiewicza 30, 30-059 Krakow, Poland

**Keywords:** memristors, spike timing-dependent plasticity, neuromorphic engineering, electrodeposition, copper tungstate, copper molybdate

## Abstract

Memristors possess non-volatile memory, adjusting their electrical resistance to the current that flows through them and allowing switching between high and low conducting states. This technology could find applications in fields such as IT, consumer electronics, computing, sensors, and medicine. In this paper, we report successful electrodeposition of thin-film materials consisting of copper tungstate and copper molybdate (CuWO_4_ and Cu_3_Mo_2_O_9_), which showed notable memristive properties. Material characterisation was performed with techniques such as XRD, XPS, and SEM. The electrodeposited materials exhibited the ability to switch between low and high resistive states during varied cyclic scans and short-term impulses. The retention time of these switched states was also explored. Using these materials, the effects seen in biological systems, specifically spike timing-dependent plasticity, were simulated, being based on analogue operation of the memristors to achieve multiple conductivity states. Bio-inspired simulations performed directly on the material could possibly offer energy and time savings for classical computations. Memristors could be crucial for the advancement of high-efficiency, low-energy neuromorphic electronic devices and technologies in the future.

## 1. Introduction

The increasing integration of computers into our daily lives greatly enhances research on information processing technologies. There are two main directions in this area: software development and hardware design. Topics which can be almost exclusively associated with the former approach, such as pattern recognition, data mining, or time series forecasting, are typically realised based on machine learning principles with the use of classic, conventional silicon-based architectures working in binary fashion [1]. The effectiveness of these techniques is undeniable, given that all major IT companies invest in deep learning research, which involves algorithms that use deep (multilayer) neural networks [2]. However, there are fundamental limits imposed on the efficiency of such purely software solutions in terms of the parallelisation of fundamental computation processes and the energy consumption/dissipation balance [2], and there exist some restrictions of a more pragmatic character (e.g., economical ones) as well [3]. For that reason, numerous attempts have been made to shift some of the workload from the software domain to the hardware domain, which catalysed research on intelligent materials capable of preprocessing information or performing “in-memory” computing [4,5,6,7].

Neuromorphic engineering offers an interesting solution to this challenge [5]. This approach focusses on the design of electronic components capable of emulation of the analogue characteristics of biological structures, primarily neurons and synapses, along with the various learning and memory processes observed in natural neural networks [8]. Such a strategy could pave the way for improving machine learning algorithms, especially when it is directly integrated into specialised hardware that exhibits synaptic plasticity and supports multiple logic states. Memristors stand out as potential foundational elements for these systems due to their inherent memory and the ability to integrate them into larger networks [9].

The memristor, the fourth passive circuit element predicted theoretically by Chua in 1970 [10] and realised experimentally in 2008 by Strukov et al. [11], is a component characterised by two conductivity states (the low resistive state (LRS) and high resistive state (HRS)) that can be gradually switched with electrical stimuli, namely an electric charge [11,12]. As classic determinants of the memristive character, three fingerprints are given: a characteristic pinched I-V hysteresis loop, a frequency-dependent operation, and the retention of states [13]. Numerous mechanisms that explain memristive behaviour can be found in the literature. The authors postulate that the ionic or vacancy [14], the formation of conductive filaments [15], tunnelling effects [16], the formation of trap states in the bulk or at interfaces [17], and other effects may be dominant, depending on the given molecular architecture [18]. The interested reader is referred to a wide range of review articles that describe the current state of memristor studies [4,12,19,20,21,22,23].

Generally, memristive devices are capable of simulating the learning rules described in Hebb’s theory of synaptic plasticity [8,24,25,26,27]. According to it, the connections between neurons, through which the neuron’s action potential moves in the direction of information flow, are strengthened, while those through which it moves in the opposite direction are weakened [28]. This is the main assumption of the spike timing-dependent plasticity (STDP) model of synaptic plasticity, which depends on the sequence of impulses occurring in the pre- and postsynaptic neuron. The strengthening and weakening of connections between neurons in the STDP model is summarised by the following saying: “Cells that fire together wire together and those who fire out of sync lose their link”. The achieved strengthening or weakening of synaptic connections is realised by long-term potentiation (LTP) and long-term depression.

Scientific work on the memristive properties present in CuWO_4_ phases served as an inspiration to continue research on mixed tungstate phases [29]. To date, there has been a lack of work in the literature that describes the memristive properties of copper molybdates and mixed copper tungstate and molybdate phases. On the other hand, various tungstate and molybdate phases find diverse applications in photocatalysis [30] and for luminescence [31,32] (e.g., for scintillator-based radiation detectors). Furthermore, their crystal structure suggests the possibility of ionic conductivity and vacancy migration, and therefore, they should be suitable for memristive applications. Therefore, an updated methodology for the synthesis of mixed copper molybdate-coper tungstate phases has been developed, and the memristive properties were investigated. Due to various existing technologies for fabrication of molybdate and tungstate materials for other applications, implementation of synaptic devices should be less technologically challenging than for other materials with still unknown large-scale fabrication protocols.

In this work, success was achieved in the synthesis of thin-film materials from the copper tungstate and molybdate (CuWO_4_, Cu_3_Mo_2_O_9_, and mixed phases) by electrodeposition on conductive glass. X-ray diffractograms (XRDs) confirmed the presence of tungstate and molybdate phases. Furthermore, X-ray photoemmision spectroscopy (XPS) was used for analysis of the concentration of particular atoms. The materials showed promising memristive properties for the −5 V–+4.2 V potential window. The switching between ON and OFF states at cyclic scans (with different frequencies) and during short pulse excitations was recorded. The thin-film materials showed a two- to threefold difference between the recorded currents for the HRS ones relative to the LRS ones. The retention time of the switched conductive states was also investigated, and the LRS persisted for up to several hours after switching. In the next step, the materials were used to simulate the STDP effect. The STDP effect was recorded for each of the studied materials. The measurements were repeated several times to ensure repeatability of the results obtained.

## 2. Materials and Methods

The electrodeposition solution was prepared by dissolving Na_2_WO_4_·2H_2_O (Acros Organics, Waltham, MA USA) in water (40 mM). Dilute nitric acid was then added to lower the pH of the solution from 9 to 4 before adding Cu(NO_3_)_2_·3H_2_O (Sigma Aldrich, Burlington, MA, USA); final concentration: 40 mM). After the addition of Cu(NO_3_)_2_·3H_2_O, the pH of the solution was adjusted to 2.75, and then the solution was heated to 70 °C before adding p-benzoquinone (Sigma Aldrich; final concentration: 50 mM). For the preparation of other materials in the series containing molybdate anion, the procedure was identical, but Na_2_MoO_4_·2H_2_O (Acros Organics) substrate was used along with (or beside) Na_2_WO_4_, with a 10 mM step (40:0, 30:10, 20:20, 10:30, and 40:0 mM ratios) for the following samples. The applied electrodeposition conditions were based on the parameters reported by Hill et al. [33]. Electrodeposition was carried out in a glass three-electrode electrochemical cell using a leakless Ag/AgCl reference electrode, platinum gauze as a counter electrode, and fluorine-doped tin oxide (FTO) glass as a working electrode (FTO glass substrate, Ossila, Sheffield, UK), which was washed with detergent, isopropanol, and deionised water and treated with oxygen plasma before being used. An SP-200 potentiostat (Bio-Logic, Seyssinet-Pariset, France) was used for the electrodepositions, and a constant potential of −0.1 V (vs. Ag/AgCl) was applied to reduce the *p*-benzoquinone (*p*-BQ) to hydroquinone (*p*-BQH_2_) until a charge of 0.72 C/cm^2^ was passed. All layers were amorphous after deposition and transformed into crystalline electrodes by annealing at 500 °C in air for 1 h (heating/cooling ramp 2 °C/min). Electrochemical reduction of benzoquinone is a process that indirectly controls the solubility product of the final products by locally controlling the pH of the electrolyte according to the reaction
*p*-BQ + 2 H_2_O + 2e^−^ → *p*-BQH_2_ + 2 OH^−^

Generation of hydroxide ions in this reaction is directly responsible for the precipitation of acid-soluble molybdate and tungstate phases. Thus, despite the lack of an electrode reaction directly responsible for formation of the desired product, this indirect electrodeposition process is highly convenient.

The XRDs were measured on a Panalytical Empyrean X-ray diffractometer, with CuKα radiation of a wavelength l = 1.78901 Å. The fitting of the diffractograms was performed with HighScore Plus software, version 2.3.

XPS measurements were performed on a PHI VersaProbeII scanning XPS system using monochromatic Al Kα X-rays (1486.6 eV) focused on a 100 µm spot and scanned over a 400 µm × 400 µm area. The photoelectron start angle was 45°, and the analyser’s transition energy was 117.50 eV (0.5 eV step) for the overview scans and 46.95 eV (0.1 eV step) to obtain high-energy resolution spectra for the C 1s, O 1s, Na 1s, Mo 3d, W 4f, and Cu 2p regions. To keep the surface potential of the sample constant regardless of its conductivity, a double charge compensation beam was used with Ar+ ions at 7 eV and electrons at 1 eV. All XPS spectra were referenced to the peak of nonfunctional saturated carbon (C-C) C1s at 285.0 eV. The pressure in the analytical chamber was maintained at less than 3 × 10^−9^ mbar. Deconvolution of the spectra was performed using PHI MultiPak software (v.9.9.0.8). The background of the spectrum was subtracted using the Shirley method.

Scanning electron microscopy (SEM) images were taken with SEM/FIB Quanta 3D (FEI, Hillsboro, OR, USA) with 5 kV HV. Four pictures were taken for each sample with 1 k and 10 k magnifications. The supporting energy dispersion spectroscopy (EDS) spectra were measured with SEM Versa 3D (FEI).

Silver electrodes 1 mm^2^ in area were sputtered onto each sample (a metal mask was used to deposit electrodes of a given area) with a EM ACE600 (Leica Microsystems GmbH, Wetzlar, Germany) high-vacuum DC magnetron sputter. The sample thickness and electrode thickness were measured with the DektakXT (Bruker, Billerica, MA, USA) stylus profilometer.

The voltammetric measurements were performed with an SP-150 potentiostat (BioLogic, France). The cyclic voltammetry (CV) was measured in a 5–4.25 V voltage window with a series of scan rates of 500 mV/s. In Appendix A, we presented CVs recorded at more differing scan rates (100–5000 mV/s). Those additional scans served as an exploratory role for the optimal scan rate and voltage window presented in this publication. Impulse measurements were performed with a standard procedure for memristive material testing, with the impulse sequence consisting of WRITE, READ, and RESET voltage pulses.

Spike timing-dependent plasticity (STDP) experiments were conducted in an ambient atmosphere using the 4200-SCS analyser (Keithley, Solon OH, USA) system in a two-electrode set-up. In this set-up, Ag was used as the working electrode, while FTO was used as the counter electrode. The device state was read three times at −0.5 V to determine its memristive state. To study the STDP, 2.5 V symmetric sawtooth signals were applied to both the working (pre-synaptic) and counter (post-synaptic) electrodes. These signals were given at intervals ranging from 20 to 400 ms. After these signals, the state was measured again using three −0.5 V pulses. The change in synaptic weight was calculated using the following formula:$$∆w=\frac{i_{post}-i_{pre}}{i_{pre}}$$
where *i_pre_* and *i_post_* are the currents measured before and after applying the sawtooth voltage signals, respectively. After the STDP tests, the device was reset using an I–V scan. For more details on the voltage patterns, refer to Appendix A.

## 3. Results

In this study, the convention for the naming of samples was adopted in relation to the concentrations of substrates and is presented in Table 1. Precise values of the atomic concentrations in the film were taken from the XPS measurements (more details are provided in the text).

### 3.1. X-ray Diffraction

The results of the XRD measurements are shown in Figure 1, where a close-up of the interesting angular range of the diffraction patterns is shown. All samples were highly crystalline with low impurity contributions. A distorted structure of tungstate, which is characteristic for CuWO_4_, can be observed for all compounds in the series (except for the compound in which tungstate was not present). Apart from the first and last compound in the series, mixed amounts of copper and molybdenum oxide compounds can be observed in the remaining samples. The first notable difference in comparison with the work of Hill et al. [33] is the lack of CuMoO_4_ phases and mixed solid solutions (like CuW_0.55_Mo_0.45_O_4_). In turn, the best fit was found for the Cu_3_Mo_2_O_9_ phases. This indicates that mixed-phase materials were obtained. The peak observed at around 39 degrees can be attributed to the FTO glass substrate.

### 3.2. X-ray Photoelectron Spectroscopy

The surface concentrations of the chemical bonds obtained from the fitting of XPS data for all analysed samples are listed in Table 2 (groups) and Table 3 (oxidation states). Table 2 and Table 3 are a single result and were divided for better presentation in the text. Survey scans as well as high-resolution scans on which the following analysis is based are placed in Appendix A.

The Mo 3d spectra were fitted with one doublet structure (d5/2–d3/2 doublet separation equals 3.13 eV) with the first main 3d5/2 line centred at 232.1 eV, which indicates MoO_3_ or Mo_2_O_5_ [35]. The W 4f spectra were fitted with a doublet structure (f7/2–f5/2 doublet separation equals 2.12 eV) with the main 4f7/2 line centred at 35.2 eV, which indicates W(VI) oxidation of tungsten, similar to that of WO_3_ [36].

The spectrum collected in the Cu 2p3/2 region was fitted with the seven components with the first line centred at 931.6 eV, which indicates the existence of a metallic state of copper, the second line centred at 932.7 eV, indicating the Cu^+^ oxidation state, and a line at 934.3 eV and 936.3 eV, which originated from the presence of the Cu^2+^ oxidation state [37,38]. Three lines within the energy range of 940–945 eV were due to the shake-up processes that were mostly intense only in the presence of the Cu^2+^ oxidation state.

The Na 1s spectra were fitted with a single line centred at 1071.2 eV, which indicates the presence of Na^+^ as the only impurity [36].

The O 1s spectrum was fitted with three components: the first main line centred at 530.2 eV, which points out the existence of metal oxides (lattice oxygen), the second line centred at 531.5, indicating the presence of either defective oxygen in metal oxides or organic species (O=C) from contamination, and the last line found at 532.8 eV that originated either from -OH- or C-O-type bonds from organic contamination [36,38].

The C 1s spectra could be fitted with three components: aliphatic carbon (285.0 eV), C-O groups (286.3 eV), and C=O or O-C=O groups, as evidenced by the line centred at 288.7 eV [39]. All the detected lines are typical of organic contamination present on the surface of air-handled samples. The C1 and O1 spectra indicate that the surface was mostly terminated with oxygen atoms of tungstate/molybdate anions, with a small contribution from absorbed carbon dioxide from the atmosphere. The presence of hydroxyl groups also cannot be ruled out.

### 3.3. Scanning Electron Microscopy

SEM images were taken for the surface characterisation of the obtained samples and are shown in Figure 2. As can be seen, all samples presented a highly porous surface. Apart from the CW sample, which was relatively uniform, the samples containing molybdate anions had cracks, aggregates of different sizes, and bubbles on their surfaces. However, the morphology of the obtained samples was consistent with that of the source work. To exclude the risk of breaking the continuity of the layer with holes to the conductive surface, additional EDS measurements were made, and even in the cracks, assumed elements could be found (figures are placed in Appendix A).

### 3.4. Electrical Measurements

In order to perform electrical measurements, the silver electrodes were sputtered first on the surface of the samples. Silver is a suitable material because its redox potential is placed in the band gap of the studied semiconductors. The thickness of the samples as well as the silver electrodes were measured and are presented in Table 4.

To inspect the memristive fingerprints, CV, impulse, and retention measurements were performed. The results of these measurements are presented in the following figures (Figure 3, Figure 4, Figure 5 and Figure 6). All the studied samples exhibited a pinched hysteresis loop, one of the key memristor fingerprints (Figure 3). Sample CWM1 showed the highest ratio of LRS to HRS currents. CW and CWM2 were also characterised by a relatively high LRS-to-HRS ratio but presented more unstable currents during cycling. Samples CWM3 and CM, although stable, showed the smallest ratio of switched states. In Appendix A, where other scan rates are included, the CWM1 sample showed the best cycling stability in relation to other samples. The sample CW, without a molybdate phase, showed the lowest current values during cycling, whereas sample CM without a tungstate phase showed the largest values for the currents during cycling. Interestingly, two samples, CWM2 and CWM3, exhibited different scan routes, as depicted by the arrows in Figure 3. This means that those samples started from LRS, and during cycling, they were switched to HRS, which is the opposite of the rest of the samples.

In the next step, impulse measurements were performed to test the repeatability of switching during short-pulse stimulation (Figure 4). The CW sample showed the best switching during the pulse test, whereas the other samples showed rather poor operation. Sample CWM2 presented mostly good switching, apart from several points from the HRS, which were near the LRS. CWM3 and CM presented variable operation, but some separation of states was still achieved, whereas sample CWM2 quickly descended into the LRS (after around 50 cycles). The impulse results showed a striking difference in relation to the CV results; in the samples where CV scans showed promising operation, the impulse measurements presented rather poor switching. This shows that the copper tungstate and molybdate samples worked better when the voltage stimulation was gradually increased and not in a short impulse manner.

On the basis of the current-voltage characteristics, an average resistance of each sample was determined (Table 4), as well as the specific resistivity (assuming the average contact surface area to be 2 mm^2^). A systematic decrease in the specific resistivity with an increasing molybdate content may be observed (with a minor discrepancy for CWM2, with the resistivity being slightly higher than for CWM1).

Finally, the retention of the switched LRS was measured in an impulse manner (Figure 5). In the duration of the test (10 h), the CW and CM samples showed relatively the best retention of the LRS, with a sharp decline in the current amplitudes after the fifth and sixth hours, respectively. Sample CWM2 showed only 2.5 h of retention, whereas the CWM1 sample showed a steep decline from the beginning of the measurement. Sample CWM3 exhibited variable responses throughout the measurement time, showing no retention at all and oscillating around the current values of the HRS.

### 3.5. Spike Timing-Dependent Plasticity

In the last step, STDP simulations were performed with the use of the studied materials. The working electrode (memristive material deposited on FTO) acted as the presynaptic connection, while the counter electrode (sputtered silver) acted as the postsynaptic connection. The results presented (Figure 6) consist of the average values of Δ*w* from among three experiments (performed one after the other for the same counter electrode). Before the start of each measurement series, the system was reset using CV scans.

In the results presented, the area where synaptic weighting was enhanced occurred for the case where the impulses reached the electrode treated as a postsynaptic electrode after they reached the presynaptic electrode. When the timing of the impulses shown in the graph took on negative values—the impulse reached the first of the postsynaptic electrodes—a weakening of synaptic weights could be observed. This is a classic case of STDP, as observed for memristive oxide materials and in biological structures. The smallest values for the standard deviations occurred for the CW and CWM2 samples. The least pronounced STDP effect was recorded for CWM1 and CWM3. The most outlined LTP case can be observed for the CWM2 sample, where for positive values of Δ*t*, there was a distinctive area of enhancement of the synaptic weights Δ*w*. For most cases, a transition region can be observed between attenuation and amplification of Δ*w*, occurring for values of Δ*t* of approximately 0–150 ms, similar to the case of biological samples. In the CWM2 sample, the transition region occurred for Δ*t* values in the range of approximately −150–0 ms. Notable is the scale of the effect, as the synaptic weights underwent a maximum 20% increase in the CWM2 sample, but with the CM sample, although more inconsistent, the synaptic weights presented a 200% increase with small Δ*t* values.

## 4. Discussion

Electrodeposition was successfully used to synthesise thin-film copper tungstate and molybdate materials on FTO conductive glass. The results for material characterisation (XRD and XPS) presented in the previous section confirm the obtained pure, crystalline samples with assumed atomic compositions. Rather small discrepancies were found in the exact atomic ratios of W:Mo in the relation values reported in the work of Hill et al. [33]. The difference in the stoichiometric composition observed in the diffraction peaks remains puzzling, despite strict adherence to the preparation procedure. In addition, the SEM images of the surfaces of the tested samples are consistent with those observed by Hill et al. [33], which may not be obvious due to the use of different magnifications.

The studied materials exhibited promising memristive characteristics within a voltage window from −5 V to +4.2 V. The voltage window was different than that reported by Sun et al. (±1.5 V) [29], but it could be attributed to different synthesis routes. The findings revealed switching between an LRS and HRS during cyclic scans of varying frequencies and during brief impulse excitations. These thin films demonstrated a significant (two- to threefold) difference in the recorded current values for the HRS compared with the LRS. However, discrepancies were noticed between the CV and impulse results, suggesting that the copper tungstate and molybdate samples performed optimally when the voltage stimulation was increased gradually rather than in short impulses. These results also show that promising CV results do not necessarily translate to good impulse operation. This may be due to the underlying mechanism of the observed memristive properties. It may be assumed that, most probably, oxygen vacancy migration is a key factor in producing memristive effects in studied materials, as it is the most commonly postulated mechanism present in various oxide materials. In filament-based or phase change-based materials, switching between the LRS and HRS occurs in an abrupt manner (sometimes called “digital”, referring to the binary nature of the switching), whereas in the presented samples, more gradual (also called “analogue”) changes with a changing voltage were observed. This digital switching occurs when a given filament breaks through a material structure into the opposite side of the device or when the material changes its phase to a more conducting one. Furthermore, memristive devices based on oxide materials may possess rather poor response times to electrical stimulation, as the oxygen vacancies migrate between the crystallites of the material, which in turn may explain the observed results for impulse voltage cycling. During CV scans, oxygen vacancies may create gradual percolation paths, whereas in impulse tests, those same vacancies may take more random paths. However, one can argue that percolation paths created by the oxygen vacancies may resemble filaments created in other memristor types. While this may be true, their stability will not be comparable, as the lattice vibration will be more likely to break percolation paths than metal filaments. The retention time of the switched LRS lasted up to a maximum of several hours after the switching for several samples. During that time, relaxation of the oxygen vacancies across the material structure probably occurred due to crystal lattice vibrations. There was variability in the retention capabilities across the samples, with some showing sharp declines in their current amplitudes after certain timeframes. Rather poor retention times exclude those materials as possible candidates for new data storage devices, belonging to the group called Resistive RAM (ReRAM) [40]. However, the retention of states was sufficient to perform STDP measurements.

One of the significant highlights of this work was the possibility that these materials can simulate the STDP effect, which is a crucial learning mechanism in biological neural systems. This effect was observed for every material in the series studied and was repeated multiple times to ensure statistical accuracy. A classic STDP case was observed, indicating synaptic weight adjustments based on the timing of the impulses. A transition region was identified between the attenuation and amplification of synaptic weights, reminiscent of observations in biological samples.

Simulating STDP in memristor devices paves the way for power-efficient neuromorphic architectures. These architectures, which are inherently parallel like the brain, promote on-chip learning, minimising external processor communications and thereby cutting down on latency and power associated with data transfers. Such integration facilitates the processing of vast amounts of data in parallel and adapts dynamically based on temporal input patterns, allowing AI systems to learn and adjust without exhaustive retraining. By mirroring biological processes, these systems foster the creation of novel algorithms inspired by neural functions, which could excel over conventional machine learning methods, especially in handling temporal datasets.

The obtained tungstate and molybdate composite materials significantly differed in electrical conductivity and memristive performance. Pure copper molybdate is the most conductive material, with a specific resistivity of only circa 80 Ω·m. Increasing the content of the tungstate phase leads to a gradual increase in the specific resistivity, and materials containing 30–50 atomic % of tungstate are about 4 times more resistive. The most resistive one is pure copper tungstate (circa 3.7 kΩ·m). Differences in the conductivity of polycrystalline thin films are a consequence of different crystal structures for the studied materials (Figure 7). Apparently, less ordered copper molybdate offers 3D chains of apex-sharing octahedrons, which may be a better arrangement in the case of polycrystalline samples. In contrast, the layered tungstate could be a better conductor but only in one dimension.

On the basis of differences in performance for the pure and mixed-phase memristive materials, it can be concluded that pure copper tungstate presents a clear and well-defined hysteresis loop, mimicking the antisymmetric Hebbian learning rule in the pulsed experiments with a 100% increase in synaptic weight and presenting the most reproducible LRS and HRS switching. First, this may be related to the purity of the crystal phases, which should result in the formation of more stable conductivity patterns. However, this does not explain the relatively poor performance of the Cu_3_Mo_2_O_9_ phase (an incredibly narrow hysteresis loop, unstable switching, and poorly performed Hebbian learning). The comparison of the crystal structures of both materials may give some hints (Figure 7). The copper tungstate phase (CuWO_4_) forms lattice apex-sharing CuO_6_ and WO_6_ octahedra, forming individual planes. Contrary to the Cu_3_Mo_2_O_9_ structure, the CuO_6_ and MoO_6_ octahedra form a complex 3D network without well-defined copper- and molybdenum-rich planes. This may make the formation of conductive pathways more difficult, and the resulting pathways may be much less stable. It does not manifest directly in the dynamic features (see Figure 6) but can be clearly seen in the switching reproducibility tests (see Figure 4).

Differences in the stoichiometry, symmetry, and lattice parameters of these structures not only exclude the possibility of solid solutions but also may be responsible for an increased number of defects in mixed samples, which is common in oxometallate phases [43].

In summary, the copper tungstate and molybdate mixed-phase materials synthesised through electrodeposition show immense potential in mimicking synaptic functionalities and memristive properties. This could bridge the gap between electronic devices and biological systems, offering new paths in the development of neuromorphic devices. Further research is needed for optimising the materials under consideration, particularly with regard to their thickness, electrode dimensions, and surface morphology. While these materials already demonstrated the capability to simulate STDP, there is ample potential to enhance their performance. Given their semiconductor nature, future studies could also explore the use of light as a modulator for STDP, offering a promising avenue for fine-tuning their properties and functionalities.

## Figures and Tables

**Figure 1 materials-16-06675-f001:**
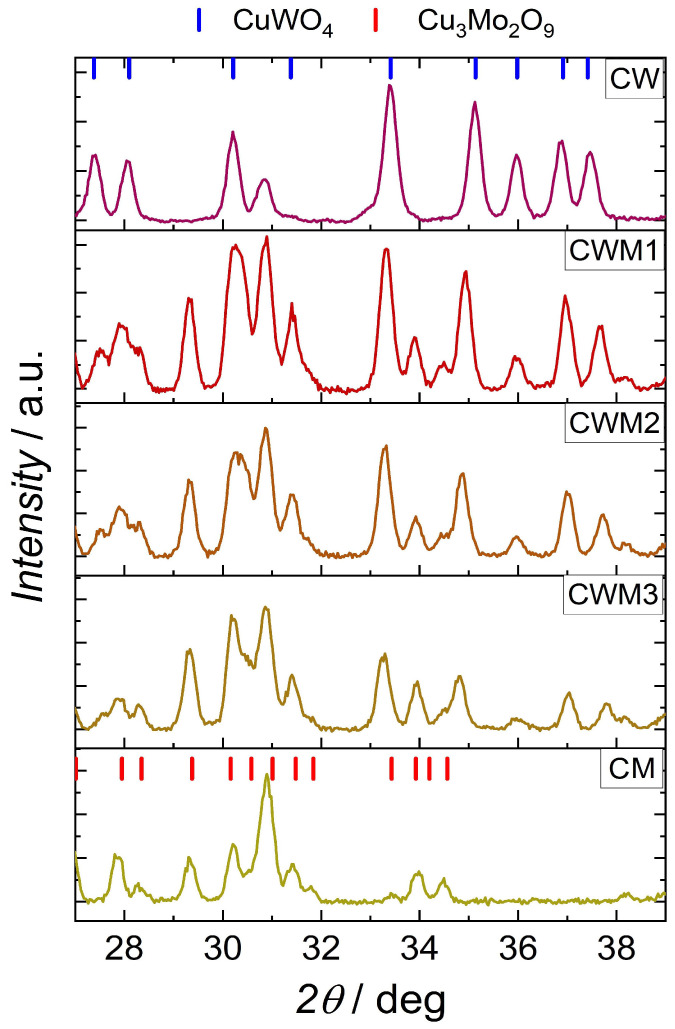
XRD spectra recorded for synthesised samples. CuWO_4_ (PDF 01-080-5325) and Cu_3_Mo_2_O_9_ (PDF 00-024-0055) reference cards presented the best fit for the obtained samples. The peak at around 39 degrees, marked with an asterisk, can be attributed to the FTO substrate glass [34]. The showed spectra are for CW, CWM1, CWM2, CWM3, and CM samples.

**Figure 2 materials-16-06675-f002:**
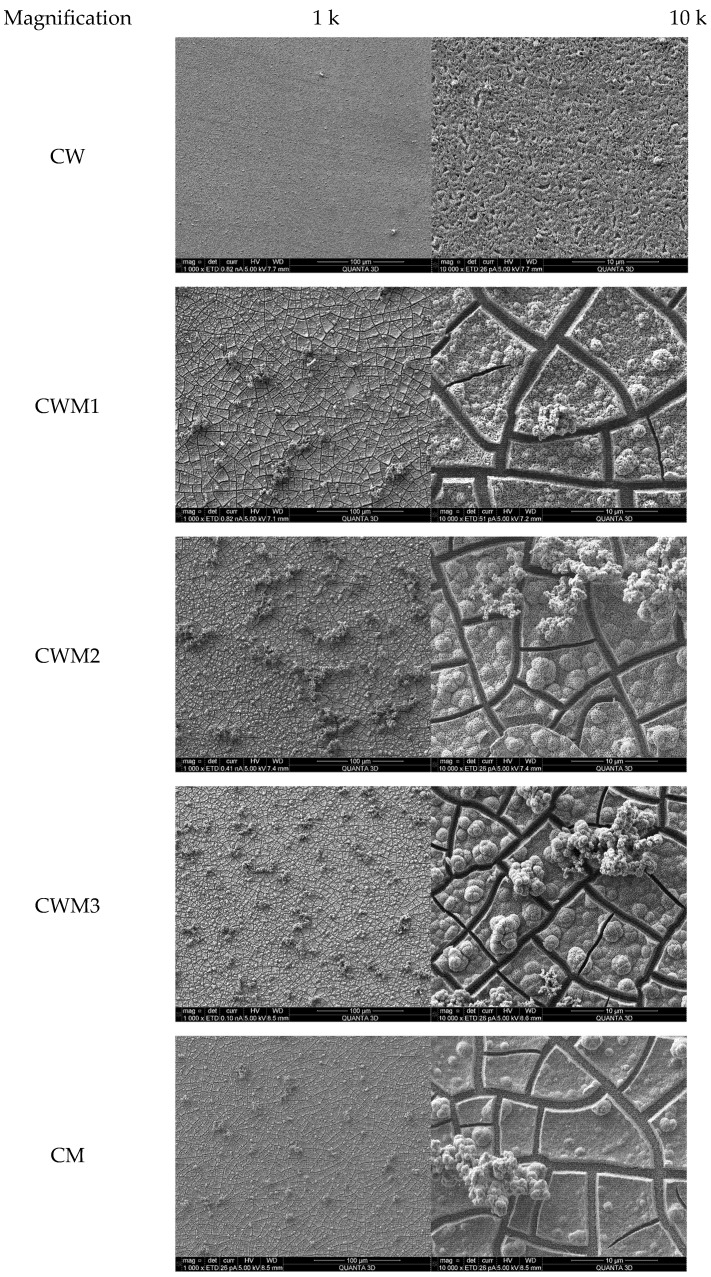
SEM images of electrodeposited samples.

**Figure 3 materials-16-06675-f003:**
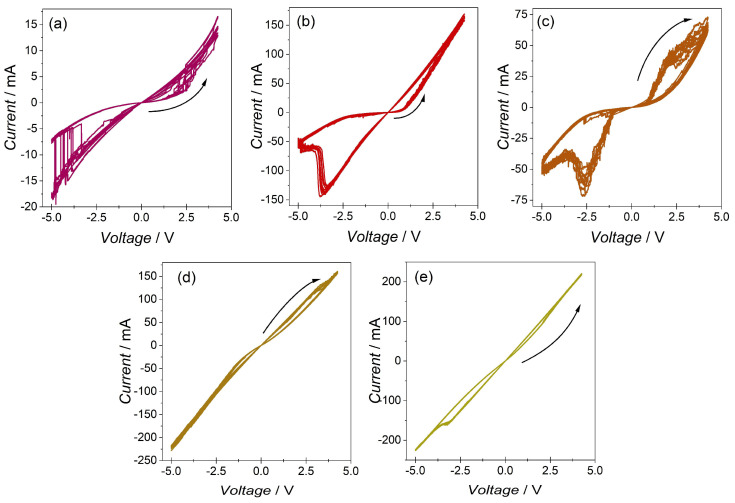
Cyclic voltammetry results for (**a**) CW, (**b**) CWM1, (**c**) CWM2, (**d**) CWM3, and (**e**) CM samples (υ = 500 mV/s). Arrows represent the directions of the scans.

**Figure 4 materials-16-06675-f004:**
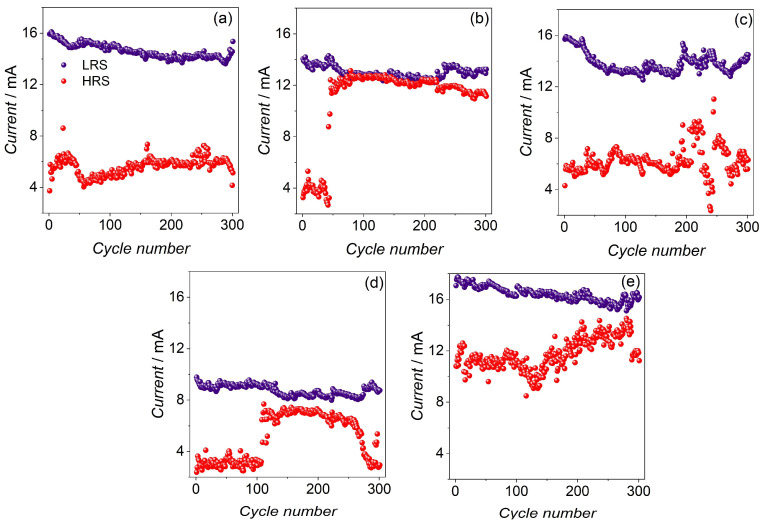
Results of the voltage impulse measurements for (**a**) CW, (**b**) CWM1, (**c**) CWM2, (**d**) CWM3, and (**e**) CM samples. LRS and HRS depict low resistive state and high resistive state, respectively.

**Figure 5 materials-16-06675-f005:**
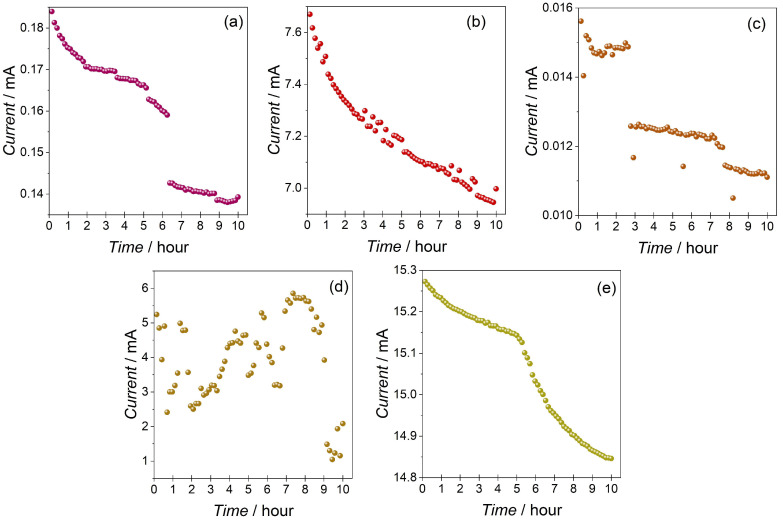
Retention of LRS impulse measurement results for (**a**) CW, (**b**) CWM1, (**c**) CWM2, (**d**) CWM3, and (**e**) CM samples. Measurements were made every 8 min for 10 h with READ voltage (−0.5 V).

**Figure 6 materials-16-06675-f006:**
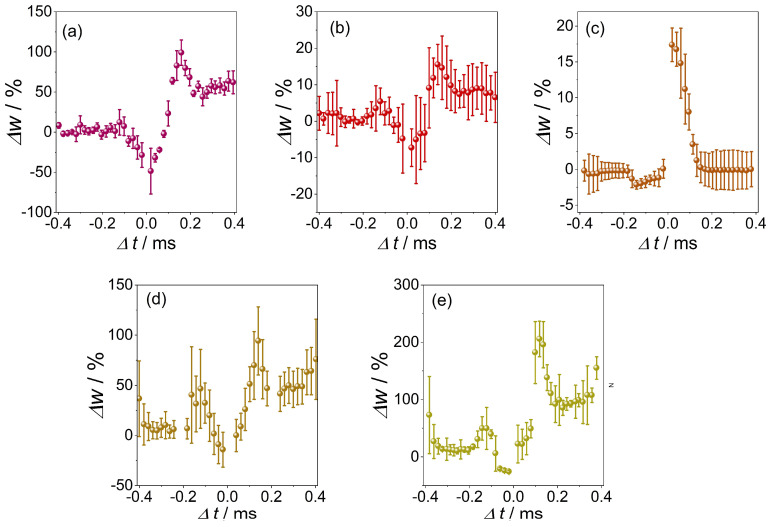
Spike timing-dependent plasticity measurements results for (**a**) CW, (**b**) CWM1, (**c**) CWM2, (**d**) CWM3, and (**e**) CM samples. The results were averaged from three different sets of measurements.

**Figure 7 materials-16-06675-f007:**
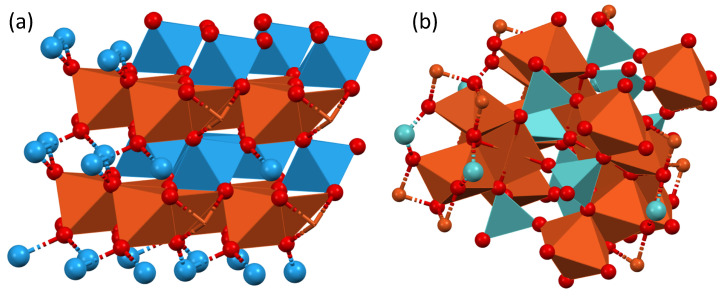
Crystal structures of CuWO_4_ (**a**) and Cu_3_Mo_2_O_9_ (**b**). Packing diagrams generated from cif files using Mercury2022.3.0. Data taken from [41] (CuWO_4_) and [42] (Cu_3_Mo_2_O_9_). Copper octahedra are orange-brown, tungstate ions are blue, and molybdate greenish. Red spheres indicate oxygen atoms.

**Table 1 materials-16-06675-t001:** Composition of the electrolytes used for the electrodeposition of particular samples and final composition of electrodeposited films.

Sample	Composition of Electrolyte		Composition in Film
	Na_2_WO_4_ (mM)	Na_2_MoO_4_ (mM)	(W:M Atomic Ratio)
CW	40	0	1.00:0.00
CWM1	30	10	0.52:0.48
CWM2	20	20	0.36:0.64
CWM3	10	30	0.20:0.80
CM	0	40	0.00:1.00

**Table 2 materials-16-06675-t002:** Surface composition (atomic %, with an error of ±0.1 atomic %) determined by fitting XPS spectra for all samples analysed.

Peak	C1s.cf1	C1s.cf2	C1s.cf3	O1s.cf1	O1s.cf2	O1s.cf3
Energy [eV]	285.0	286.3	288.7	530.2	531.5	532.8
Groups	C-C	C-O	C=O, O-C=O	MeOx (Crystalline)	MeOx (def.) O=C	O-C-OH
CW	13.2	2.6	0.8	48.6	6.6	1.6
CWM1	11.5	2.6	0.6	49.4	5.8	1.5
CWM2	12.5	2.0	0.8	47.0	8.0	1.6
CWM3	12.4	3.2	0.4	48.9	6.7	1.8
CM	16.8	2.3	1.5	46.2	4.6	1.4

**Table 3 materials-16-06675-t003:** Surface composition (atomic %, with an error of ±0.1 atomic %) determined by fitting XPS spectra for all samples analysed.

Peak	Na1s.cf1	Cu2p3.cf1	Cu2p3.cf2	Cu^2+^	Mo3d.cf1	W4f.cf1
Energy (eV)	1071.2	931.6	932.7	934.3	232.1	35.2
Oxidation States	Na^+^	Cu^0^	Cu^+^	Cu^2+^	Mo^6+^	W^6+^
CW	1.7	1.9	0.0	6.5	0.0	16.4
CWM1	1.9	4.3	2.0	4.9	7.0	8.6
CWM2	2.4	2.6	0.7	6.8	9.5	6.2
CWM3	2.1	2.0	0.7	6.4	11.9	3.4
CM	1.7	2.3	1.5	6.7	14.8	0.0

**Table 4 materials-16-06675-t004:** Thicknesses of the thin films and of the sputtered silver electrodes of the studied samples.

Sample	*d_layer_* (nm) ^a^	*d_electrode_* (nm) ^b^	*R* (kΩ)	*σ* (Ω·m)
CW	630 ± 50	50 ± 7	464.0 ± 0.1	3682.8 ± 0.2
CWM1	1000 ± 50	63 ± 7	43.3 ± 0.1	216.7 ± 0.2
CWM2	1600 ± 50	57 ± 7	92.9 ± 0.1	290.2 ± 0.2
CWM3	1300 ± 50	80 ± 7	24.9 ± 0.1	95.9 ± 0.2
CM	1320 ± 50	53 ± 7	21.3 ± 0.1	80.7 ± 0.2

^a^ Thickness of the layer. ^b^ Thickness of the electrode.

## Data Availability

Data supporting the findings of this study are available from the corresponding author upon reasonable request. Electrical measurement data are available as ASCI as well as instrument-specific binary files.

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
