# Peer review of "The Memristive Properties and Spike Timing-Dependent Plasticity in Electrodeposited Copper Tungstates and Molybdates"

_materials, 2023, doi:10.3390/ma16206675_

Round 1
Reviewer 1 Report
This manuscript presents a study on electrodeposited copper tungstate and molybdate oxides for memristor applications. The results indicate successful deposition of copper tungstate and molybdate oxides and their mixed phases. CV measurements were performed to demonstrate the memristive functions of the material. The paper is well written, however there are only limited results to support the memristor characteristics and the STDP. Therefor the manuscript should be revised before publication in materials.
1. The introduction discusses too much details about the memristors and STDP in a very advanced level, but the results obtained from the study are not detailed enough.
2. What is the novelty in this work, it is not well presented in the introduction. What is the rationale for choosing these particular materials.
3. The details of electrodeposition is missing in the manuscript. What is the role of p-benzoquinone in the experiments?
4. How much quantity of impurities are identified in the electrodeposited film? As it is a wet chemical method there might be some impurities which could ruin the electrical performance.
Author Response
Authors truly appreciate insightful comments. The revised manuscript has beed corrected and all of the comments have been taken into account.
- The introduction discusses too much details about the memristors and STDP in a very advanced level, but the results obtained from the study are not detailed enough.
In out opinion the introduction shows the broader context of the research and may be useful for all readers for very familier with prospective applications of memristors and other synaptic devicers, therefore we would prefer not to shorten the introductory part.
2. What is the novelty in this work, it is not well presented in the introduction. What is the rationale for choosing these particular materials.
The novelty of the work (first current-drived copper molybdate and tungstate memristors) as well as the rationale (well known fabrication and physics of these materials) is highlighted in the manuscript.
3. The details of electrodeposition is missing in the manuscript. What is the role of p-benzoquinone in the experiments?
The role of benzoquinone is described in full detail, which also shows the mechanism of electrodeposition.
4. How much quantity of impurities are identified in the electrodeposited film? As it is a wet chemical method there might be some impurities which could ruin the electrical performance.
The only impurity found in materials were surface carbon dioxide and organic contamination from atmosphere, as well as minor quantities of sodium ions, occluded from the electrolyte.
Reviewer 2 Report
Reviewer Recommendation and Comments for manuscript materials-2620319 with the title: “Memristive properties and spike-timing dependent plasticity in electrodeposited copper tungstate and molybdate oxides”, authors: D. Przyczyna, K. Mech, E. Kowalewska, M. Marzec, T. Mazur, P. Zawal, K. Szaciłowski.
The authors present the electrodeposition of copper tungstate and copper molybdate (CuWO4 and Cu3Mo2O9) thin films with memristive properties. Electrodeposits are characterized physico-chemically using techniques such as XRD, XPS and SEM.
The main comments that I find useful for improving the quality of the article are presented below:
# Are the terms "copper tungstate oxide and copper molybdate oxides" used correctly? For example, copper tungstate oxide = copper tungsten oxide = copper tungstate? The authors use these terms interchangeably. Perhaps only one name should be used. Same observation for both title and the entire article.
# If possible, the authors should introduce some citations regarding the use of these materials in medicine. Do copper, tungsten and molybdenum raise toxicity problems or not?
# The template must be used / citations in the text.
In the text, reference numbers should be placed in square brackets [ ] and placed before the punctuation; for example [1], [1–3] or [1,3]. For embedded citations in the text with pagination, use both parentheses and brackets to indicate the reference number and page numbers; for example [5] (p. 10), or [6] (pp. 101–105).
# line 96. “Cu(NO3)2 ‧ 3H2O” Underscript must be used.
# line 99. “molybdate oxide anion”. Molybdate anion?
# line 124. “3 ·10-9mbar”. The typo must be corrected.
# line 143. “Experiments”. The typo must be corrected.
# line 171. “CuW0.55Mo0.45O4”. The typo must be corrected.
# Figure 1 should be moved to section 3.1. X-Ray diffraction
# Does the working solution used for electrodeposition also contain p-benzoquinone? Can the C peaks be attributed to the presence of this organic compound?
# Table 3. Why does the CM sample show a peak for W6+ (0.1)?
# Tables 2 and 3. Are there periods or commas?
# line 230. “(Figure 7-9).” ?
# line 235. “In ESI (Section X)”. There is no Section X in ESI!
# The clarity/pixels of the figures need to be improved, especially for figures 3-6.
# The authors should introduce discussions about the differences in recorded currents. From the lowest (15 mA for CW) to the highest (200 mA for CM). Similar for figures 4, 5 and 6.
# There are some grammar and typing mistakes.
# The authors must revise the entire manuscript.
Moderate editing of English language required
Author Response
Authors thank Reviewer for very careful check-up of the manuscript and insighful comments. All of them have been takes into acocunt in the revised version.
# Are the terms "copper tungstate oxide and copper molybdate oxides" used correctly? For example, copper tungstate oxide = copper tungsten oxide = copper tungstate? The authors use these terms interchangeably. Perhaps only one name should be used. Same observation for both title and the entire article.
Nomenclature of thin films is unified and corrected in the revised namuscript.
# If possible, the authors should introduce some citations regarding the use of these materials in medicine. Do copper, tungsten and molybdenum raise toxicity problems or not?
There are no special precautions when handlung this materials. Their form (thin ceramic films) is much safer than nanoparticles, therefore in opur opinion no discussion on toxicity is needed in this manuscript.
# The template must be used / citations in the text.
In the text, reference numbers should be placed in square brackets [ ] and placed before the punctuation; for example [1], [1–3] or [1,3]. For embedded citations in the text with pagination, use both parentheses and brackets to indicate the reference number and page numbers; for example [5] (p. 10), or [6] (pp. 101–105).
All references and punctuation are corrected in the reivces manuscript.
# line 96. “Cu(NO3)2 ‧ 3H2O” Underscript must be used.
Corrected.
# line 99. “molybdate oxide anion”. Molybdate anion?
Corrected.
# line 124. “3 ·10-9mbar”. The typo must be corrected.
Corrected.
# line 143. “Experiments”. The typo must be corrected.
Corrected.
# line 171. “CuW0.55Mo0.45O4”. The typo must be corrected.
Corrected.
# Figure 1 should be moved to section 3.1. X-Ray diffraction
Corrected - figure moved as close to the description as possible.
# Does the working solution used for electrodeposition also contain p-benzoquinone? Can the C peaks be attributed to the presence of this organic compound?
The role of benzoquinone is described in detail. In our opinion, due to thermal treatment there si no possibility of benzoquinone contributing to the inpurities of films.
# Table 3. Why does the CM sample show a peak for W6+ (0.1)?
This was a typping error - corrected.
# Tables 2 and 3. Are there periods or commas?
Corrected.
# line 230. “(Figure 7-9).” ?
Corrected.
# line 235. “In ESI (Section X)”. There is no Section X in ESI!
Corrected.
# The clarity/pixels of the figures need to be improved, especially for figures 3-6.
Corrected.
# The authors should introduce discussions about the differences in recorded currents. From the lowest (15 mA for CW) to the highest (200 mA for CM). Similar for figures 4, 5 and 6.
A short discussion on condictivity of layers, taking into account the variation of film thickness is added to the revised manuscript.
# There are some grammar and typing mistakes.
DONE.
# The authors must revise the entire manuscript.
DONE
Reviewer 3 Report
This is a very good paper, but in its current form the article is not yet ready for publication and requires some improvements and clarification.
1. In the introduction it would be very useful to add more background information about tungstates (including CuWO4) and their applications. This would make it possible to arouse greater interest among readers working with such and similar materials. It is known that materials from this family works as a detector at CERN, they are important for medical diagnostic purposes, or photocatalysis.
Raizada, P., et al (2020). Performance improvement strategies of CuWO4 photocatalyst for hydrogen generation and pollutant degradation. Journal of Environmental Chemical Engineering, 8(5), 104230.
Mikhailik, V. B., et al (2005). Luminescence of CaWO4, CaMoO4, and ZnWO4 scintillating crystals under different excitations. Journal of Applied Physics, 97(8).
Millers, D., et al (1997). The temperature dependence of scintillation parameters in PbWO4 crystals. physica status solidi (b), 203(2), 585-589.
2. Tables 2 and 3. There should be a dot instead of a comma. What is the measurement error there? If possible, error bars need to be added.
3. The same for determining film thickness.
4. What can be said about the surface termination of the films?
5. Fig.3-6. Small details in the drawings are difficult to discern. Needs improvement.
Author Response
Authors thank the Reviewer for insightful comments. All of them have been taken into accound when revising the manuscript.
1. In the introduction it would be very useful to add more background information about tungstates (including CuWO4) and their applications. This would make it possible to arouse greater interest among readers working with such and similar materials. It is known that materials from this family works as a detector at CERN, they are important for medical diagnostic purposes, or photocatalysis.
Raizada, P., et al (2020). Performance improvement strategies of CuWO4 photocatalyst for hydrogen generation and pollutant degradation. Journal of Environmental Chemical Engineering, 8(5), 104230.
Mikhailik, V. B., et al (2005). Luminescence of CaWO4, CaMoO4, and ZnWO4 scintillating crystals under different excitations. Journal of Applied Physics, 97(8).
Millers, D., et al (1997). The temperature dependence of scintillation parameters in PbWO4 crystals. physica status solidi (b), 203(2), 585-589.
All papers cited as requested, some additional discussion added to the introduction.
2. Tables 2 and 3. There should be a dot instead of a comma. What is the measurement error there? If possible, error bars need to be added.
Measurement errors added to Table cations as requested.
3. The same for determining film thickness.
Info added as required.
4. What can be said about the surface termination of the films?
Comment on film purity and termination is added to section of XPS spectroscopy.
5. Fig.3-6. Small details in the drawings are difficult to discern. Needs improvement.
Corrected as requested.
Round 2
Reviewer 1 Report
This article could be considered for publication in 'materials'.
Reviewer 3 Report
Аfter successful revision this manuscript can be recommended for publication.